# Gender-Specific Prevalence and Risk Factors of Sarcopenic Obesity in the Korean Elderly Population: A Nationwide Cross-Sectional Study

**DOI:** 10.3390/ijerph20021140

**Published:** 2023-01-09

**Authors:** Jongseok Hwang, Soonjee Park

**Affiliations:** 1Institute of Human Ecology, Yeungnam University, Gyeongsan 38541, Republic of Korea; 2Department of Clothing and Fashion, Yeungnam University, Gyeongsan 38541, Republic of Korea

**Keywords:** sarcopenic obesity, risk factors, prevalence, odds ratio, the elderly

## Abstract

Sarcopenic obesity (SO) is characterized by the combination of sarcopenia and obesity. This study evaluates the gender-specific prevalence of SO and examines the identified gender-specific risk factors in a community-dwelling elderly population aged 75–84 years. A total of 813 subjects participated in the study via the Korea National Health and Nutrition Examination Survey. The SO prevalence in males and females was 15.46% (95%CI: 11.36–20.70) and 13.59% (95%CI: 10.59–17.28), respectively. The clinical sex-specific risk factors for males were low height, high weight, body mass index, waist circumference, skeletal muscle index, fasting glucose, and triglycerides. The clinical risk factors for females were low height, high weight, body mass index, waist circumference, skeletal muscle index, smoking status, fasting glucose, total cholesterol, and systolic blood pressure. These results are essential to assist healthcare professionals and primary care clinicians with early detection, diagnosis, and intervention for potential SO patients by acknowledging the sex-based prevalence and risk factors.

## 1. Introduction

Sarcopenic obesity (SO) is defined as the coexistence of both sarcopenia and obesity [1]. Sarcopenia is a condition characterized by age-related loss of muscle mass, strength, and/or physical function. Obesity is characterized by excess fat mass accumulation, which has a negative effect on health [2].

The proportion of the elderly population in Asia is rapidly increasing, and Korea is one of the most aged nations in Asia. The elderly population in Korea is expected to soar to 40% by 2050 from 15% of the population in 2022, growing to 19 million from the current 5 million [3]. Thus, age-related complications, such as sarcopenia, are a potential threat to Korean society in the future. Furthermore, the prevalence of obesity in Korea has continuously increased from 2009 to 2019, from 34.1% to 40.2% for males in their 70s and from 37.6% to 42.9% for females in their 70s. The elderly population in their 80s showed a higher prevalence in 2019 as well, with prevalence increasing from 32.0% to 40.3% in males and 28.4% to 35.4% in females [4]. As the obese population among the elderly increases, and the elderly population itself increases, the potential risk factors for SO in Korean society are also increasing. Therefore, the importance of SO prevention in Korean society is becoming essential.

The combination of obesity and low muscle mass aggravates health function, promotes the development of chronic degenerative diseases and disability, and increases the duration of long-term hospitalization. Prolonged long-term hospitalization is a great burden for the sustainability and efficiency of health initiatives in primary and secondary care, social support, public health, and policymaking [5]. Several studies have shown that individuals with SO have worse morbidity, disability, and mortality than those with low muscle mass or obesity [6,7]. Moreover, obesity hampers the generation or maintenance of muscle mass, making its diagnosis and the identification of its clinical consequences a challenge [8].

Despite the importance and critical consequences of SO, healthcare professionals and primary care clinicians lack the knowledge and diagnostic tools to diagnose SO. The average general practitioner spends less than 10 min per patient visit. For a primary care clinician to consider making a referral for the diagnosis and treatment of SO, they must recognize that there is a likelihood that the individual may have it [9]. The average clinician’s lack of knowledge about the existence of sarcopenia as a disease further increases the likelihood that the diagnosis will be missed [10]. Knowledge of the key characteristics of risk factors related to early detection and prevention is crucial [11]. The early diagnosis of SO depends on detecting symptomatic patients as early as possible. Once an SO diagnosis and intervention is delayed or missed, greater problems result associated with poor functional recovery treatment, poor prognosis, decreased quality of life, and a waste of the government health care budget.

Most SO studies have classified research participants into a single group, despite the level of physical activity being highly related to the blood test results, body composition, and health traits of the older population, which vary according to age. Thus, classifying older people by age group is fundamental to accurately evaluate the characteristics of SO. The elderly population should be divided into young-old (65–74), old (75–84), and oldest-old (85+ years) groups [3,9,12,13,14].

Additionally, there is a discrepancy among past sex-specific SO prevalence studies [15,16,17,18]. Dufour et al. [16] and Bouchard et al. [15] reported a higher prevalence of SO in men, whereas Kim et al. [17] and Stenholm et al. [18] reported a higher prevalence in women. Furthermore, there is no consensus on sex-specific risk factors in previous SO research on the elderly.

The present study evaluated the sex-specific prevalence of SO and identified the sex-specific risk factors in the community-dwelling elderly population aged 75 to 84 years. This study assumed the following two hypotheses: First, there is a sex-specific prevalence of sarcopenia in community-dwelling older populations. Second, sex-based risk factors exist in the elderly population with sarcopenia.

## 2. Materials and Methods

### 2.1. Study Participants, Study Design, and Ethical Consideration

The Ministry of Health and Welfare and the Korea Institute for Health and Social Affairs conducted the Korea National Health and Nutrition Examination Surveys (KNHANES) annually. This research is based on a total of 37,573 participants collected from the KNHANES from 2008 to 2011. The research excluded 35,431 subjects who were not 75 to 84 years old; thus, the remaining 2142 subjects were included in the research. Additionally, 1329 participants were excluded because they did not undergo the DEXA examination and did not respond to the health survey. Finally, 813 subjects were included in the analysis of elderly individuals aged 75–84 years. A total of 134 participants were assigned to the SO group, and the remaining 679 were assigned to the normal group. The SO group’s inclusion criteria were (1) age between 75 and 84 years and (2) the coexistence of both sarcopenia and obesity. The normal group’s inclusion criteria were (1) age between 75 and 84 years and (2) the existence of neither sarcopenia nor obesity. Table 1 presents the clinical characteristics of the participants. A cross-sectional design was used in this study, and the institutional review board of the Centers for Disease Control and Prevention approved the present study. The approval numbers are as follows: 2008-04EXP-01-C, 2009-01CON-03-2C, 2010-02CON-21-C, and 2011-02CON-06-C. All research subjects agreed to participate and signed an informed consent form.

### 2.2. Classification of SO

The two constituents of SO, sarcopenia and obesity, are defined as follows: First, sarcopenia is defined as low skeletal muscle mass determined by measuring the sum of the appendicular skeletal muscle mass (ASM). ASM is measured and calculated using dual X-ray absorptiometry (QDR4500A; Hologic, Inc., Bedford, MA, USA). The skeletal muscle mass index (SMI) is calculated as ASM (kg) divided by body mass index (BMI) (kg/m^2^). The Foundation for the National Institutes of Health Sarcopenia Project in the United States developed the criteria for sarcopenia classification. The recommended SMI cutoff for sarcopenia is <0.789 in men and <0.521 in women [19].

Second, obesity is defined as abnormal or extensive fat accumulation that negatively affects health and is characterized by a BMI ≥ 25 kg/m^2^ and central obesity as a waist circumference (WC) greater than 90 cm in males and 80 cm in females among the Asian population [20].

### 2.3. Data Collection

Demographic data included age, sex, height (cm), weight (kg), BMI, WC, SMI, fasting glucose, triglycerides, total cholesterol, systolic blood pressure (SBP), diastolic blood pressure (DBP), smoking status, and drinking status. WC is gauged by passing a midpoint between the basement of the rib cage and the uppermost of the lateral border of the iliac crest during full exhalation. Laboratory blood tests were measured after eight hours of fasting. Alcohol drinkers and cigarette smokers were sorted into non-users, ex-users, and current users.

### 2.4. Statistical Analyses

Data analysis was conducted using SPSS version 22.0 for Windows (SPSS Inc., Chicago, IL, USA). We applied weight values reflecting the entire nationwide elderly Korean population. Each sample weight was assigned to each individual in the following sequence: (1) calculation of base weight, (2) adjustment for non-responses, and (3) post-stratification adjustment to match the entire previous census population. All analyses were performed using complex sampling analysis. Each individual adjusted the weights provided by KNHANES. Independent *t*-tests were performed for parametric variables, and chi-square tests were used for non-parametric variables, comparing the variables between the SO and normal groups. Multiple logistic regression with adjusted covariates was performed to predict SO and to determine the odds ratio value of the SO risk factor in males and females. The alpha value was designated as 0.05 for statistical significance with all the variables.

## 3. Results

### 3.1. Sex-Specific Prevalence in SO

The weighted values of SO prevalence in males and females were 15.46% (95%CI: 11.36–20.70) and 13.59% (95%CI: 10.59–17.28), respectively. The prevalence of SO was higher in males than in females (Table 2).

### 3.2. Clinical Risk Factors in Males

SO risk factors in males included height, weight, BMI, WC, SMI, FG, and triglyceride level, showing statistical significance (*p* < 0.05). In contrast, age, smoking status, drinking status, total cholesterol level, SBP, and DBP were not significant (*p* > 0.05). Table 3 lists the sex-specific clinical parameters associated with SO.

### 3.3. Clinical Risk Factors in Females

The significant risk factors for SO in women were height, weight, BMI, WC, SMI, smoking status, FG, TC, and SBP (*p* < 0.05). However, age, triglyceride level, and DBP did not show a statistically significant difference (*p* > 0.05) (Table 3).

### 3.4. Odds Ratio for SO in Males

Odds ratios were calculated using multiple logistic regression analysis. The following variables were statistically significant: BMI, WC, SMI, FG, and triglyceride level (*p* < 0.05). The respective odds ratios for each were 5.942 (1.862–18.587), 6.018 (4.693–7.716), 0.255 (0.242–0.269), 0.987 (0.957–1.000), and 0.938 (0.931–0.946). Height and weight differences were not statistically significant (*p* > 0.05). The odds ratio for SO in males is described in Table 4.

### 3.5. Odds Ratio for SO in Females

Height, weight, BMI, WC, SMI, FG, TC, and SBP variables were statistically significant (*p* < 0.01). Their respective odds ratios were 0.038 (0.003–5.133), 1.197 (0.005–27.530), 2.001 (0.150–26.550), 11.291 (8.515–14.972), 0.330 (0.315–0.345), 1.153 (1.131–1.176), 1.044 (1.017–1.072), and 0.837 (0.795–0.881) (Table 5).

## 4. Discussion

This study investigated sex-specific SO prevalence and its risk factors in the elderly population aged 75–84 years. Regarding sex-specific prevalence, the prevalence in males was 15.46% (11.36–20.70), which was higher than that in females, at 13.59% (10.59–17.28). This result is in line with previous studies in Canada and the US [15,16]. Defour et al. [16] conducted a Framingham cohort study of 767 elderly people in the US with an average age of 79 years and reported a prevalence rate of 8% in men and 4% in women. Likewise, Bouchard et al. [15] investigated 904 elderly Canadians aged 68–82 years, concluding that the male and female incidence rates of SO were 19% and 11%, respectively.

One plausible underlying mechanism for the greater SO incidence rates in males is associated with the hormone levels of insulin-like growth factor-1 and testosterone [21,22,23,24,25]. Females lose more muscle mass and muscle power faster than males at the initial stage of aging as a result of menopause [24,26,27]. However, in the later stages of aging, a decline in insulin-like growth factor-1 and testosterone hormone levels in males aggravates the loss of muscle function and mass at a high speed. The redundant calories from low energy expenditure due to the loss of skeletal muscle and function increase the propensity of obesity in males, which ultimately contributes to SO in males [23,25].

Regarding the sex-specific clinical parameters associated with SO, fasting glucose was highly related to SO in both sexes. This finding is in line with those of several studies [28,29,30]. Perna et al. [28] analyzed 639 patients with an average age of 80.90 years and found that the SO group had higher glycemia. Similarly, Lu et al. [30] investigated 600 community-dwelling elderly individuals and concluded that the SO group had higher fasting blood glucose levels than the normal and pure sarcopenia groups. Du et al. [29] assessed 631 community-dwelling elderly individuals in East China and reported that those with SO showed greater blood glycemia than the normal population.

One possible underlying mechanism is that skeletal muscle is essential for postprandial glucose regulation. Approximately 80% of insulin-dependent glucose uptake is located in the muscles after food is absorbed from the stomach and intestines. The process of insulin-dependent and -independent skeletal muscle glucose production necessitates glucose transport from the circulation to the muscle. The insulin-dependent and -independent glucose cross the cell membrane via the extracellular matrix and translocate to the cell membrane during exercise, secreting insulin. The glucose gradient promotes uptake through the catalyzed glucose transporters, and glucose transport is regulated by intracellular glucose metabolism [31]. A diminished skeletal muscle glucose uptake after ingestion is associated with abnormal carbohydrate metabolism, causing high blood glucose levels.

Another risk factor for SO is the smoking status in females. This result is consistent with those of previous risk factor studies in patients with SO [32,33]. Atkins et al. conducted a cohort study in an elderly SO population and found that smoking is a risk factor compared with the normal elderly population. Androga et al. [32] examined 11,616 people in the US, finding that more people in the SO group (59.1%) had experience with tobacco than the normal population (48.1%). One possible underlying mechanism for smoking as a risk factor is that cigarette smoking aggravates skeletal muscle loss by suppressing muscle protein synthesis, promoting muscle catabolism, and accumulating visceral fat. First, the fractional synthesis rate of muscle decreased in smokers, and genes such as E3 ubiquitin ligase, muscle atrophy F-box, and muscle growth inhibitor were found in smokers [34]. The type I fiber atrophy, increased glycolytic capacity, and reduced expression of constitutive nitric oxide synthases found in smokers reduce the volume of skeletal muscle [35]. In particular, smoking in females reduces estrogen levels, a hormone that is highly related to muscle synthesis [36]. Female smokers have higher testosterone [37] and lower estrogen levels in the blood [38]. Furthermore, smoking helps in gaining waist and visceral fat. This is because smoke changes high-density lipoprotein cholesterol, plasma glucose, immunoreactive insulin, and insulin resistance [39]. Thus, smoking helps reduce muscle mass and promotes visceral fat in female SO.

Triglycerides are a risk factor in males, a finding that is consistent with previous SO studies [29,30,40]. An SO study in China reported that only men had significantly higher triglycerides (2.40 mmol/L) than the normal elderly population (1.60 mmol/L) [29], while the change in females was not significant. A Korean longitudinal study [40] described that the triglyceride level in male SO (149.7 mg/dL) surpassed that in the normal group (98.3 mg/dL). Lu et al. [30] investigated 600 community-dwelling elderly and reported significant changes between the SO and normal groups, with 1.9 mmol/L and 1.3 mmol/L, respectively. 

Total cholesterol is a risk factor for female elderly patients with SO and several studies corroborated our results [28,29,30].

A cross-sectional study of SO in eastern China [29] revealed that the total cholesterol level in female SO group participants was higher than that in the normal group. Lu et al. [30] reported that the cholesterol levels in the SO group were higher than those in the normal group. Perna et al. [28] investigated more than 600 elderly Italian subjects and reported that the SO group had greater total cholesterol in the blood than the healthy group. 

A plausible rationale for such high values of triglycerides and total cholesterol relates to insulin resistance [41] and a high volume of inflammatory cytokines [42].

Our SBP findings support the notion that SBP is a risk factor for the elderly female population found in previous studies [30,32,33]. Lu et al. [30] examined SO in the elderly population in Taiwan, noting that SBP in the SO group was 131.6 mmHg and greater (126.8 mmHg in the normal population). Atkins et al. [33] conducted a British cohort study of elderly males and found that the SO group was significantly higher than the normal group. Several possible underlying reasons have been revealed for why muscle loss is interdependent with metabolic alterations and is associated with a decline in energy expenditure and physical inactivity, causing insulin resistance and arterial stiffness in the elderly [43,44,45]. Increased visceral fat mass leads to an inflammatory reaction, thickens the walls of blood vessels, obstructs the flow of blood, and narrows the vascular passages [46]. Females, in particular, have lower skeletal muscle and higher adipose tissue than males [47], making women more susceptible to hypertension. This combination of low muscle mass and accumulated adipose tissue in the visceral area may result in higher SBP in the female SO population.

The clear findings of the present study provide the first clinical evidence related to sex-specific prevalence and clinical risk factors for SO in the elderly population, using research data that are representative of the Korean population, and the gold standard measurement including DEXA for the evaluation of sarcopenia. Nevertheless, this study still has shortcomings that need to be considered in future research. Due to the nature of the cross-sectional study design, our findings may be more credible using a longitudinal design, with measurements across different timelines with the same elderly individuals. In addition, the prevalence of SO was not fully dealt with in comparison with other studies in Korea and other countries. The study could be more beneficial if it investigated other national prevalences of SO.

## 5. Conclusions

This study provides the first clinical evidence of the sex-specific prevalence and clinical risk factors in the Korean elderly with SO. The prevalence of SO in males was higher than that in females, and the weighted values of SO prevalence in males and females were 15.46% (95%CI: 11.36–20.70) and 13.59% (95%CI: 10.59–17.28), respectively. The clinical risk factors for SO in females were low height, high weight, BMI, WC, SMI, smoking status, FG, TC, and SBP. For males, the clinical risk factors were low height, high weight, BMI, WC, SMI, FG, and triglyceride levels. Acknowledging the sex-specific risk factors and prevalence for healthcare professionals and primary care clinicians is essential for early detection, diagnosis, and intervention for potential SO patients.

## Figures and Tables

**Table 1 ijerph-20-01140-t001:** General characteristics of the participants (*n* = 813).

Variables	Sarcopenia Obesity (*n* = 134)	Normal (*n* = 679)	*p*
Gender (men/women) (%)	39.52/60.45	40.06/59.94	0.843
Age (years)	77.857 ± 1.875	77.821 ± 1.918	0.839
Height (cm)	153.58 ± 8.99	155.743 ± 9.089	0.011
Weight (kg)	66.476 ± 8.173	53.567 ± 9.627	0.000 **
BMI (kg/m^2^)	28.173 ± 2.538	22.020 ± 2.939	0.000 **
WC (cm)	96.768 ± 5.036	79.556 ± 9.506	0.000 **
SMI (kg/m^2^)	0.562 ± 0.139	0.703 ± 0.150	0.000 **
Smoking status (%) (current-/ex-/non-smoker)	22.14/15.27/62.59	30.16/13.05/56.78	0.168
Drinking status (%) (current-/ex-/non-smoker)	31.82/27.27/40.91	39.27/34.60/26.13	0.227
FG (mg/dL)	114.3 ± 28.618	101.443 ± 26.116	0.000 **
Triglyceride (mg/dL)	164.613 ± 88.004	132.168 ± 84.504	0.000 **
TC (mg/dL)	198.874 ± 41.554	189.034 ± 35.77	0.009
SBP (mmHg)	138.060 ± 18.916	133.202 ± 18.017	0.004
DBP (mmHg)	75.992 ± 9.47	74.559 ± 10.247	0.131

Values are mean ± standard deviation. The independent *t*-test and chi-square test were significant at *p* < 0.01 **. BMI, body mass index; WC, waist circumference; SMI, skeletal muscle index; FG, fasting glucose; TC, total cholesterol; SBP, systolic blood pressure; DBP, diastolic blood pressure.

**Table 2 ijerph-20-01140-t002:** Prevalence of sex-specific sarcopenic obesity.

	Males (*n* = 325)			Females (*n* = 487)		
	Sarcopenic Obesity (*n* = 53)	Normal (*n* = 272)	Total	Sarcopenic Obesity (*n* = 81)	Normal (*n* = 406)	Total
Un-weighted (%)	16.31	83.69	100	16.63	83.37	100
Weighted (%)	15.46 (11.36–20.70)	84.54 (79.30–88.64)	100	13.59 (10.59–17.28)	86.41 (82.72–89.41)	100

Weighted values represent the 95% confidence interval.

**Table 3 ijerph-20-01140-t003:** Sex-specific clinical parameters associated with sarcopenic obesity (75–84).

	Males			Females		
	Sarcopenic Obesity (*n* = 53)	Normal (*n* = 272)	*p*	Sarcopenic Obesity (*n* = 81)	Normal (*n* = 406)	*p*
Age (years)	77.731 ± 1.941	77.618 ± 1.933	0.698	77.938 ± 1.839	77.946 ± 1.899	0.973
Height (cm)	163.148 ± 4.417	165.128 ± 5.028	0.008 **	147.437 ± 4.817	149.943 ± 5.468	0.000 **
Weight (kg)	72.035 ± 6.228	58.853 ± 9.086	0.000 **	62.907 ± 7.246	50.301 ± 8.434	0.000 **
BMI (kg/m^2^)	27.027 ± 1.51	21.531 ± 2.816	0.000 **	28.909 ± 2.788	22.289 ± 2.979	0.000 **
WC (cm)	96.800 ± 4.685	80.545 ± 9.778	0.000 **	96.747 ± 5.277	78.941 ± 9.289	0.000 **
SMI (kg/m^2^)	0.726 ± 0.048	0.877 ± 0.069	0.000 **	0.457 ± 0.040	0.596 ± 0.057	0.000 **
Smoking status (%) (current-/ex-/non-smoker)	49.02/25.49/25.49	55.41/29.18/15.40	0.205	5.00/8.75/86.25	14.26/2.89/82.85	0.004 **
Drinking status (%) (current-/ex-/non-smoker)	50.99/37.25/11.76	56.86/28.76/14.38	0.463	19.75/20.99/59.26	28.18/24.48/47.32	0.122
FG (mg/dL)	114.773 ± 25.188	98.745 ± 22.121	0.000 **	113.985 ± 30.878	103.244 ± 28.356	0.005 **
Triglyceride (mg/dL)	167 ± 92.779	114.39 ± 63.986	0.000 **	163.045 ± 85.402	144.049 ± 94.015	0.121
TC (mg/dL)	185.545 ± 30.150	178.055 ± 32.703	0.155	207.627 ± 45.710	196.371 ± 35.890	0.023 *
SBP (mmHg)	135.192 ± 17.553	130.955 ± 18.411	0.123	139.901 ± 19.626	134.586 ± 17.647	0.014 *
DBP (mmHg)	75.75 ± 9.156	74.331 ± 10.282	0.351	76.148 ± 9.720	74.700 ± 10.233	0.235

Values are mean ± standard deviation; independent *t*-test and chi-square test were significant at *p* < 0.05 *, *p* < 0.01 **. BMI, body mass index; WC, waist circumference; SMI, skeletal muscle index; FG, fasting glucose; TC, total cholesterol; SBP, systolic blood pressure; DBP, diastolic blood pressure.

**Table 4 ijerph-20-01140-t004:** Odds ratio for SO in males.

Variables	Odds Ratio (95% of CI)	*p*
Height	0.039 (0.000–6.236)	0.209
Weight	2.241 (0.004–11.492)	0.089
BMI	5.942 (1.862–18.587)	0.000 **
WC	6.018 (4.693–7.716)	0.000 **
SMI	0.255 (0.242–0.269)	0.000 **
FG	0.987 (0.957–1.000)	0.047 *
Triglyceride	0.938 (0.931–0.946)	0.000 **

All values represent multiple logistic regressions with 95% confidence intervals (CIs). Significance levels are *p* < 0.05 *, *p* < 0.01 **. BMI, body mass index; WC, waist circumference; SMI, skeletal muscle index; FG, fasting glucose.

**Table 5 ijerph-20-01140-t005:** Odds ratios for SO in females.

Variables	Odds Ratio (95% of CI)	*p*
Height	0.038 (0.003–5.133)	0.000 **
Weight	1.197 (0.005–27.530)	0.000 **
BMI	2.001 (0.150–26.550)	0.000 **
WC	11.291 (8.515–14.972)	0.000 **
SMI	0.330 (0.315–0.345)	0.000 **
FG	1.153 (1.131–1.176)	0.000 **
TC	1.044 (1.017–1.072)	0.001 **
SBP	0.837 (0.795–0.881)	0.000 **

All values represent multiple logistic regressions with 95% confidence intervals (CIs). The significance level is *p* < 0.01 **. BMI, body mass index; WC, waist circumference; SMI, skeletal muscle index; FG, fasting glucose; TC, total cholesterol; SBP, systolic blood pressure.

## Data Availability

All data were anonymized and can be downloaded from the website at https://knhanes.kdca.go.kr/knhanes (accessed on 1 January 2023).

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
