# Peer review of "Gender-Specific Prevalence and Risk Factors of Sarcopenic Obesity in the Korean Elderly Population: A Nationwide Cross-Sectional Study"

_ijerph, 2023, doi:10.3390/ijerph20021140_

Round 1
Reviewer 1 Report
The aim of the manuscript was to assess the gender-specific prevalence of sarcopenic obesity (SO) and investigate the identified gender-specific risk factors in a community-based elderly Korean population aged 75–84 years. In their results, the authors also identified gender-specific risk factors in this community of elderly population. In the introduction part: - in this section, it would be appropriate to add a few new literary sources from the past 5 years. In the material and methods section: - it is necessary to write what were the inclusion (e.g., age, waist circumference, glycemia, etc.) and exclusion criteria for the selection of samples - (e.g., oncology patients, COPD patients, diabetes mellitus type 2; insulin resistance, systemic inflammatory diseases and etc.) - this data is missing. - This research is based on a total of 37,573 participants collected by 82 of the 2008 KNHANES, from which people with SO were selected. To determine the current prevalence of OS in the population, more recent data would be appropriate, but of course for the purposes of determining sex-specific risk factors (which was one aim of the study), these data are appropriate. - in the manuscript, unify the use of the terms sarcopenic obesity and sarcopenia obesity (differences in text and tables). Results - I have no comments. Discussion - in this section, it would be appropriate to add a few new literary sources from the past 5 years. Conclusion - I have no comments. References 37 literary sources were used in the manuscript (without self-citations), of which 8 sources are from the last 5 years; 12 resources for the last 5-10 years and 17 resources that are older than 10 years. Literary sources need to be partially updated with newer ones and thus supplemented with others.
Author Response
Response to the reviewers’ comments
At first, authors express their deep gratitude to the reviewer’s valuable comments. we know it is arduous job for reviewing papers. It consumes lots of time and effort. We were able to learn the way to write paper correctly because of your delicate comments. And we came to realize citing new references is very important, again. We really appreciate it with my whole heart.
*Please find attached our revised manuscript with changed from the original version highlighted with green(Please click an author's note file).
- In the introduction part: - in this section, it would be appropriate to add a few new literary sources from the past 5 years.
Authors response: Authors totally agree with the reviewer’s comment and feel sorry for Sorry for citing old literary sauces. We update recent papers as much as possible (lines 46, 57, 58, page 1)
- In the material and methods section: - it is necessary to write what were the inclusion (e.g., age, waist circumference, glycemia, etc.) and exclusion criteria for the selection of samples - (e.g., oncology patients, COPD patients, diabetes mellitus type 2; insulin resistance, systemic inflammatory diseases and etc.) - this data is missing. –
Authors response: Authors feel sorry for missing SO group and normal group criteria . We added SO group and normal group criteria (lines 89-91, page 2, lines 108-115, page 3)
- This research is based on a total of 37,573 participants collected by 813 of the 2008 KNHANES, from which people with SO were selected. To determine the current prevalence of OS in the population, more recent data would be appropriate, but of course for the purposes of determining sex-specific risk factors (which was one aim of the study), these data are appropriate.
Authors response: Authors entirely agree with the reviewer’s comment. The reason why we use years of 2008-2011 is that muscular skeletal mass data by DEXA were included only these years. 2011-2022 data did not include DEXA experiment results.
- In the manuscript, unify the use of the terms sarcopenic obesity and sarcopenia obesity (differences in text and tables).
Authors response: Authors feel sorry for not using unified expression. We changed to sarcopenic obesity in table and text (Table 2 in page 6, Table 3 in Table 4).
- Discussion - in this section, it would be appropriate to add a few new literary sources from the past 5 years.
Authors response: Authors entirely agree with the reviewer’s comment. This was amended (lines 193, 197, 234, 256 pages 7-8)
- References 37 literary sources were used in the manuscript (without self-citations), of which 8 sources are from the last 5 years; 12 resources for the last 5-10 years and 17 resources that are older than 10 years. Literary sources need to be partially updated with newer ones and thus supplemented with others.
Authors response: Authors entirely agree with the reviewer’s comment. These were changed in comments both 1 and 5.
Authors sincerely express their gratitude to the reviewer’s valuable comments, which have improved the readability and quality of this paper a lot.

Reviewer 2 Report
Thank you for permitting me to review this manuscript
whille the objectives which are declared at the end of introduction , are sex related specific prevalence and risk factors , they do not appear in the title
The title should reflect the sex issue of the claimed objectives
ABSTRACT
The risk factors enumeration should be more precise , for ex is it low height? , high weight? etc..
methods
the selection criteria of patients is not clear please provide more details from the begining , somthing like 33000 subjects and finishing 813
results:
did the authors assessed the odd ratio of the whole group (men and women) and check withother studies this population , could it be done?
discussion
line 185 please provide reference (PPR)
It appears some difference in the incience of SO between this study and similiar previously published studies , please compare the incidence and discuss the difference in a mini chapter with other similar reports
This study provides the first clinical evidence of sex-specific prevalence and clinical 274 risk factors in the Korean elderly with SO. The prevalence of SO in males was higher than 275 that in females, and the weighted values of SO prevalence in males and females were 276 15.46% (95%CI: 11.36 - 20.70) and 13.59% (95%CI: 10.59 - 17.28), respectively
Please check the litterature again as this study may not be the first study looking for sex difference , if not please compare your results
Author Response
Response to the reviewers’ comments
At first, authors express their deep gratitude to the reviewer’s valuable comments. We know it is arduous job for review papers. It consumes lots of time and effort. We were able to learn the way to write paper correctly because of your delicate comments. We really appreciate it with my whole heart.
*Please find attached our revised manuscript with changed from the original version highlighted with Yellow (Please click an author's note file).
- While the objectives which are declared at the end of introduction, are sex related specific prevalence and risk factors, they do not appear in the title. The title should reflect the sex issue of the claimed objectives
Authors response: Authors totally agree with the reviewer’s comment and feel sorry for missing the clear explanation for “sex-specific”. We changed it to “Gender-Specific Prevalence and Risk factors of Sarcopenic Obesity in the Korean Elderly Population: A Nationwide Cross-Sectional Study” (lines 1-2, page 1).
- In ABSTRACT, the risk factors enumeration should be more precise, for ex is it low height? , high weight? etc..
Author response: Authors express their deep gratitude to the reviewer’s valuable comment. We amended it (lines 14-17, page 1).
- In methods, the selection criteria of patients is not clear please provide more details from the beginning, something like 33000 subjects and finishing 813
Authors totally agree with the reviewer’s comment. We corrected it (lines 83-89, page 2).
- In results: did the authors assessed the odd ratio of the whole group (men and women) and check with other studies this population, could it be done?
Author response: Thank you very much for the reviewer’s valuable comments.
Sure, we do examine closely all the study of these population.
First of all, we would like to mention the importance of categorized age and We examine specific age between 75-84 which called old age with this following reason.
The ages of older adults can be divided into three categories: “young-old”, “old,” and “oldest-old” The age of young-old ranges from 65 to 74 years; the old ranges from 75 to 84 years old, and the oldest-old are over 85 years of age. (Papalia, D. Human Development; McGraw-Hill Humanities Social: 2008, Kulik, C.T.; Ryan, S.; Harper, S.; George, G.J.A.o.M.J. Aging populations and management. 2014, 57, 929-935. Lee, S.B.; Oh, J.H.; Park, J.H.; Choi, S.P.; Wee, J.H. Differences in youngest-old, middle-old, and oldest-old patients who visit the emergency department. Clin Exp Emerg Med 2018, 5, 249-255, doi:10.15441/ceem.17.261. little, W. Introduction to Sociology Creative Commons Atrtribution: 2016.)
Second, there is a discrepancy in the present study and other study with the population in the prevalence. This is because of a different definition of sarcopenia. To be spefic, we applied The Foundation for the National Institutes of Health Sarcopenia Project in the United States developed the criteria(Studenski, S.A.; Peters, K.W.; Alley, D.E.; Cawthon, P.M.; McLean, R.R.; Harris, T.B.; Ferrucci, L.; Guralnik, J.M.; Fragala, M.S.; Kenny, A.M.J.J.o.G.S.A.B.S.; et al. The FNIH sarcopenia project: rationale, study description, conference recommendations, and final estimates. J. Gerontol. A Biol. Sci. Med. Sci. 2014, 69, 547-558, doi:10.1093/gerona/glu010.) The skeletal muscle mass index (SMI) is calculated as ASM (kg) divided by body mass in-dex (BMI) (kg/m2). The Foundation for the National Institutes of Health Sarcopenia Project in the United States developed the criteria for sarcopenia classification. The recommended SMI cutoff for sarcopenia is <0.789 in men and <0.521 in women
On the other hands, Kim and his colleague (Kim YS, Lee Y, Chung YS, Lee DJ, Joo NS, Hong D, Song GE, Kim HJ, Choi YJ, Kim KM. Prevalence of sarcopenia and sarcopenic obesity in the Korean population based on the Fourth Korean National Health and Nutritional Examination Surveys. Journals of Gerontology Series A: Biomedical Sciences and Medical Sciences. 2012 Oct 1;67(10):1107-13) reports sarcopenia obesity prevalence based on the other definition of sarcopenia. They used the recommendations of European working group on sarcopenia in older people.
Thus, difference prevalence rate are based on diverse sarcopenia definition.
- In discussion session, line 185 please provide reference.
Author response: Authors fully agree with the reviewer’s comment. It was changed (line 191, page 6).
- It appears some difference in the incidence of SO between this study and similar previously published studies, please compare the incidence and discuss the difference in a mini chapter with other similar reports
Author response: Authors feel sorry for missing other study prevalence rates of SO, comparing similar reports. Authors have plan to investigated prevalence of SO in Asia countries such as Korea, Japan, Taiwan, China, and Singapore. We will discuss specifically prevalence rate in the future research.
We noted on limitation part in the discussion session (lines278-281, page8)
Thank you so much for the comments.
- This study provides the first clinical evidence of sex-specific prevalence and clinical risk factors in the Korean elderly with SO. The prevalence of SO in males was higher than that in females, and the weighted values of SO prevalence in males and females were 15.46% (95%CI: 11.36 - 20.70) and 13.59% (95%CI: 10.59 - 17.28), respectively. Please check the literature again as this study may not be the first study looking for sex difference, if not please compare your results.
Authors response: Thank you very much for your suggestion. Authors have contemplated the reviewer’s comment and we would like to give the following explanation:
Most of studies investigated specific years or assessed subject only male or female subjects. (Kwon YN, Yoon SS, Lee KH. Sarcopenic obesity in elderly Korean women: a nationwide cross-sectional study. Journal of bone metabolism. 2018 Feb 28;25(1):53-8. Hwang B, Lim JY, Lee J, Choi NK, Ahn YO, Park BJ. Prevalence rate and associated factors of sarcopenic obesity in Korean elderly population. Journal of Korean medical science. 2012 Jul 1;27(7):748-55.)
Furthermore, most of them are classified subjects into a single group(Kwon YN, Yoon SS, Lee KH. Sarcopenic obesity in elderly Korean women: a nationwide cross-sectional study. Journal of bone metabolism. 2018 Feb 28;25(1):53-8, Hwang B, Lim JY, Lee J, Choi NK, Ahn YO, Park BJ. Prevalence rate and associated factors of sarcopenic obesity in Korean elderly population. Journal of Korean medical science. 2012 Jul 1;27(7):748-55.), despite the health condition and body composition of the elderly differing according to their age. However, our present study evaluated whole available date of 2008-2011, including both men male and female. This is because muscular skeletal mass data by DEXA were included only 2008-2011 years. 2011-present data did not include DEXA experiment.
Most sarcopenia studies classified subjects into a single group, despite the health condition and body composition of the elderly differing according to their age. Thus, dividing the elderly population according to age is crucial to proper investigation of the characteristics of sarcopenia. The ages of older adults can be divided into three categories: “young-old”, “old,” and “oldest-old” The age of young-old ranges from 65 to 74 years; the old ranges from 75 to 84 years old, and the oldest-old are over 85 years of age. (Papalia, D. Human Development; McGraw-Hill Humanities Social: 2008, Kulik, C.T.; Ryan, S.; Harper, S.; George, G.J.A.o.M.J. Aging populations and management. 2014, 57, 929-935. Lee, S.B.; Oh, J.H.; Park, J.H.; Choi, S.P.; Wee, J.H. Differences in youngest-old, middle-old, and oldest-old patients who visit the emergency department. Clin Exp Emerg Med 2018, 5, 249-255, doi:10.15441/ceem.17.261. little, W. Introduction to Sociology Creative Commons Atrtribution: 2016.)
And we focused on both male and female and targeting range is age between 75-84(‘Old’ age range) with sarcopenic obesity. And None of study investigated these 75 to 84 years old which categorized by above papers, although some article published with another age range. To the best of your knowledge, the present study first clinical evidence of the prevalence and clinical risk factors according to gender in the sarcopenic elderly population in Korea. It means not available in other literatures, as far as I concerned.
Authors feel sorry for misleading you by missing the clear explanation. We apologized to confuse you, again.
Thank you very much for the reviewer’s so many valuable comments. Thanks to the reviewer, the authors have improved their understanding of this research contents and the quality of the paper a lot. Once again, thank you.
